# Inducible Polarized Secretion of Exosomes in T and B Lymphocytes

**DOI:** 10.3390/ijms21072631

**Published:** 2020-04-10

**Authors:** Victor Calvo, Manuel Izquierdo

**Affiliations:** 1Departamento de Bioquímica, Instituto de Investigaciones Biomédicas Alberto Sols CSIC-UAM, Facultad de Medicina, Universidad Autónoma de Madrid, 28029 Madrid, Spain; vcalvo@iib.uam.es; 2Department of Metabolism and Cell Signalling, Instituto de Investigaciones Biomédicas Alberto Sols CSIC-UAM, 28029 Madrid, Spain

**Keywords:** exosomes, T lymphocytes, B lymphocytes, polarized secretion, immune synapse, T-cell receptor, B-cell receptor, multivesicular bodies, diacylglycerol, MHC-class II compartment

## Abstract

Exosomes are extracellular vesicles (EV) of endosomal origin (multivesicular bodies, MVB) constitutively released by many different eukaryotic cells by fusion of MVB to the plasma membrane. However, inducible exosome secretion controlled by cell surface receptors is restricted to very few cell types and a limited number of cell surface receptors. Among these, exosome secretion is induced in T lymphocytes and B lymphocytes when stimulated at the immune synapse (IS) via T-cell receptors (TCR) and B-cell receptors (BCR), respectively. IS formation by T and B lymphocytes constitutes a crucial event involved in antigen-specific, cellular, and humoral immune responses. Upon IS formation by T and B lymphocytes with antigen-presenting cells (APC), the convergence of MVB towards the microtubule organization center (MTOC), and MTOC polarization to the IS, are involved in polarized exosome secretion at the synaptic cleft. This specialized mechanism provides the immune system with a finely-tuned strategy to increase the specificity and efficiency of crucial secretory effector functions of B and T lymphocytes. As inducible exosome secretion by antigen-receptors is a critical and unique feature of the immune system this review considers the study of the traffic events leading to polarized exosome secretion at the IS and some of their biological consequences.

## 1. Introduction

### 1.1. A Brief History of Exosomes: Exosome Timeline and Relevant Facts

Exosomes are small membrane vesicles (50–150 nm) secreted by a multitude of cell types as a consequence of the fusion of multivesicular bodies (MVB) with the plasma membrane [1,2]. MVB are subcellular compartments containing intraluminal vesicles (ILV) [3,4] that are part of the endosomal system, which also includes early endosomes, late endosomes, and lysosomes [5,6]. MVB are formed by inward budding from the limiting membrane of endosomes and subsequent pinching off of budding vesicles into the luminal space of endosomes. ILV present in MVB are then released into the extracellular medium as so-called exosomes [7]. Thus, MVB are members of the endocytic pathway, which are involved in an alternative secretory pathway [6]. The term exosome was first proposed to describe the exfoliated, shedding microvesicles (40 nm up to 1000 nm) harboring 5′-nucleotidase activity present in cultures from various normal and neoplastic cell lines [8]. Subsequently, the same term was proposed to define nanovesicles (30–100 nm) of endosomal origin that are released by MVB fusion with the plasma membrane [1]. Initially described in reticulocytes as a means to extrude specific obsolete components during red cell maturation, exosomes remained minimally investigated and referenced for the following 10 years (Figure 1). However, work developed in some immunocompetent cells, such as B lymphocytes [9] and dendritic cells (DC) [10] demonstrated these cells secreted exosomes, nanovesicles of endosomal origin, which expressed functional class I and class II Major Histocompatibility Complex molecules (MHC-I and MHC-II, respectively) bound to the antigenic peptide. The fact that exosomes derived from both human and murine B lymphocytes induced antigen-specific, MHC-II-restricted T cell responses suggested a role for exosomes in antigen presentation in vivo [9,10]. An important contribution was to demonstrate that T lymphocyte activation with mitogens induced the secretion of 100–200 nm microvesicles containing pro-apoptotic FasL and Apo2L [11]. Shortly after, it was verified these microvesicles were indeed canonic exosomes, since they derived from FasL^+^Apo2L^+^ ILV upon MVB fusion with the plasma membrane [12].

These early results pointed out that exosome secretion could be induced by cell activation, and exosomes might have much wider biological functions than removal of certain unwanted proteins, such as intercellular antigen-specific immunoregulation [6], including apoptosis induction [11]. Since then, an explosion in the field of extracellular vesicles has taken place (from 2 references in 1996 up to more than 2600 in 2019, Figure 1) and a relevant proportion of these citations corresponds to immunocompetent cells. An important discovery was to show mast cells-secreted exosomes contain both mRNA and microRNA (miRNA), which can be delivered to another recipient cell and can be functional changing recipient cell behavior [13]. Another major milestone in the exosome field was the discovery that stimulation of certain cell surface receptors triggered inducible exosome secretion in mast cells, cytotoxic T lymphocytes (CTL), helper T lymphocytes (Th) and B lymphocytes (reviewed in [14]). In addition, these exosomes have an important role in intercellular communication during crucial immune effector responses such as antigen presentation and apoptosis [14]. In this context, a major contribution was to formally demonstrate that inducible Th and B lymphocyte exosome secretion occurred in a polarized manner at the IS between Th lymphocytes and APC [15] or B and Th lymphocytes [16], although early studies of target cell-CTL synapses pointed out in this direction [17]. Moreover, the finding that miRNA-containing, T lymphocyte-derived exosomes may transfer genetic information to recipient APC at the IS [18,19] extended to the Th IS the previous findings establishing exosomes as a novel mechanism of genetic exchange between cells [13].

More recently, exosomes have been considered as just one type of EV. The International Society of Extracellular Vesicles (ISEV, https://www.isev.org/) endorses “extracellular vesicle” as the “generic term for particles naturally released from the cell that are delimited by a lipid bilayer and cannot replicate, i.e., do not contain a functional nucleus” [20] including exosomes (endosomal origin), ectosomes, microvesicles, microparticles (different types of plasma membrane-shedding vesicles), and apoptotic bodies. The “minimal experimental requirements for definition of extracellular vesicles” from ISEV provide recommendations on experimental methodology and minimal information on reporting EV isolation/purification, EV characterization and studies on EV biological function [21,22]. Regarding exosome biogenesis and composition please refer to the ISEV standards-curated, web pages regarding proteins, miRNAs and lipids commonly found and enriched in exosomes and other EV (EXOCARTA: http://exocarta.org/index.html# and VESICLEPEDIA: http://www.microvesicles.org/) and some excellent and recent reviews on this subject [23,24,25]. With regards to exosome heterogeneity, growing evidence supports that the same cell type releases distinct exosome subpopulations with unique compositions [26], possibly originating from different subpopulations of MVB and reflecting endosomal sorting complex required for traffic (ESCRT)-dependent and independent sorting machineries [24]. Further dissection of exosome heterogeneity requires the development of multiparameter, high-throughput analysis methods of single vesicles [27].

Taking into account these milestones and facts, exosomes have been extensively studied as biological vehicles for intercellular communication in several physiological and pathological circumstances, and indeed constitute promising biomarkers for both diagnosis and prognosis in several relevant pathologies.

### 1.2. Immune Synapse and Secretory Traffic

IS establishment by T and B lymphocytes upon TCR or BCR binding to antigen bound to APC is a very dynamic, plastic and critical event involved in antigen-specific, cellular and humoral immune responses [28,29]. The IS is described by the formation of a concentric spatial pattern, the supramolecular activation complex (SMAC), upon cortical actin reorganization [30,31,32,33]. This reorganization yields a central cluster of antigen receptors bound to an antigen called central SMAC (cSMAC) and a surrounding adhesion molecule (i.e., LFA-1)-rich ring called peripheral SMAC (pSMAC), which appears to be crucial for adhesion with the APC [28,34]. Surrounding the pSMAC, at the edge of the contact area with the APC, is the distal SMAC (dSMAC), which consists of a circular array of dense filamentous actin (F-actin) [30,35] (Figure 2). The formation of the cSMAC, pSMAC, and dSMAC characterizes a mature IS and is the basis of a signaling platform that integrates signals and coordinates molecular interactions leading to both exocytic, but also endocytic processes necessary for an appropriate antigen-specific immune response [30,36]. Actin reorganization plays a central role in IS maintenance, but also antigen receptor-derived signaling [31]. It is out of the scope of this review a detailed description of the TCR and BCR-derived signaling involved in actin cytoskeleton remodeling upon IS formation. Thus please refer to the excellent reviews on this subject [37] [33,35,38,39,40]. In brief, TCR and BCR stimulation, together with the interaction of adhesion and costimulatory molecules at the IS, triggers the generation of several early second messengers such as calcium and diacyglycerol (DAG) [41]. DAG regulates several protein kinase C (PKC) family members (including PKCδ and PKCθ) and protein kinase D (PKD1), but also a Ras/ERK2 pathway activator [42], leading to the activation of the two major actin regulatory pathways: the FMNL-1 and Dia pathway involved in F-actin nucleation, and the CDC42/WASP/ARP2/3 pathway involved in actin filament branching [43,44].

At the early stages of IS formation, F-actin accumulates at the contact area of the lymphocyte with the APC to generate filopodia and lamellipodia, that produce dynamic changes between extension and contraction in the lymphocyte over the surface of the APC [35]. Subsequently, once IS growth has stabilized, cortical F-actin accumulates into the dSMAC, and F-actin reduction from the cSMAC appears to facilitate secretion toward the APC by focusing secretion vesicles on the IS [45]. F-actin reduction at the cSMAC not simply allows secretion, as apparently plays an active role in MTOC movement to the IS [45,46]. The cSMAC is referred to as actin poor or actin hypodense region as it contains a fine F-actin network, which can only be visualized using super-resolution imaging. [40,47]. Moreover, adding more complexity to the role of cortical F-actin in secretion, recent evidence in CTL support that lytic granule fusion was not observed in zones of sustained F-actin reduction. Instead, it tended to occur at the F-actin poor base located between F-actin-rich protrusions that extend towards the target cell from central and intermediate locations within the synapse [48]. These protrusions, which depended on the cytoskeletal regulator WASP and the ARP2/3 actin nucleation complex, were required for synaptic force exertion and efficient killing [48]. Please refer to a recent review of actin remodeling at the IS for additional details [40]. Some results suggest that cortical actin reorganization at the IS is necessary and sufficient for MTOC and lytic or cytokine-containing granules polarization [46,49,50]. For instance, in T lymphocytes, dynein is recruited in a DAG/PKC-dependent manner, via the Adhesion and Degranulation Promoting Adapter Protein (ADAP) anchored to the F-actin-rich area at the dSMAC [51,52]. Dynein is a minus end-directed motor that pulls on microtubules to reorient the MTOC to the F-actin poor area at the cSMAC [50,52] (Figure 2).

The concentric F-actin architecture of the IS and the actin cytoskeleton reorganization are strikingly shared by CD4^+^ Th lymphocytes, CD8^+^ CTL, B lymphocytes, and natural killer cells (NK) [35,39]. Interestingly, all these immune cells exhibit the ability to form synapses and to directionally secrete stimulatory or cytotoxic factors at the IS. Thus, the ability to secrete these factors in a polarized manner in the context of the synaptic architecture most probably enhances the specificity and the efficacy of the subsequent responses to these factors [35].

Early consequences of IS formation and maturation are the convergence of the secretory granules towards the MTOC and the almost simultaneous polarization of the MTOC towards the cSMAC of the IS, an F-actin poor area that includes the secretory domain [29,45,53] (Figure 2). Both the convergence of secretory granules towards the MTOC and MTOC polarization to the IS appear to be necessary for optimal polarized and focused secretion at the synaptic cleft in many immune cell types capable to form synapses, including primary CD4^+^ T lymphocytes [54] and Jurkat cells [55], B lymphocytes [33], and NK cells [56,57]. This polarized secretion appears to assure the efficacy of critical effector functions of T and B lymphocytes, by reducing nonspecific cytokine-mediated stimulation of bystander cells, avoiding the killing of irrelevant target cells, favoring apoptotic suicide via activation-induced cell death (AICD) or impeding nonspecific antigen extraction from the APC. A remarkable feature of the immune system is that, whereas virtually all cell types can constitutively secrete exosomes [23], T and B lymphocytes, and mast cells are the only cell types in which surface receptor stimulation (TCR, BCR, and Fc-receptor, respectively) can induce polarized MVB traffic [37] and exosome secretion [14,15,58,59,60].

Next, we will discuss different types of existing IS and their involvement in exosome secretion.

## 2. T Lymphocyte–APC Immune Synapse: Exosome Secretion by T Lymphocytes

To constitute an IS, T lymphocytes must recognize processed antigenic peptides loaded onto MHC molecules present on the cell surface of professional APC or pathogen-infected cells. TCR interaction with peptide-MHC-I complexes (pMHC-I) induces naïve CD8^+^ CTL activation (priming), whereas TCR interaction with peptide-MHC-II complexes (pMHC-II) leads to CD4^+^ Th lymphocyte activation (Figure 2) [61]. Primed CTL form IS with target cells resulting in a specific killing. In addition, mature IS formation can induce T lymphocyte anergy or apoptosis [62].

### 2.1. CTL-Target Cell Immune Synapse

CTL-target cell IS induces the rapid polarization (from seconds to few minutes) of CTL MTOC and lytic granules (secretory granules or secretory lysosomes-SL-with MVB structure) towards the cSMAC at the IS (Figure 2). Lytic granules fusion with the plasma membrane (degranulation) induces the secretion of certain cytotoxic factors such as perforin and granzymes to the synaptic cleft, triggering target cell apoptosis [63]. Upon degranulation, FasL located at the secretory granule limiting membrane becomes exposed to the plasma membrane at the IS and induces target cell Fas crosslinking leading to target cell apoptosis [58,64,65,66].

Another consequence of degranulation is ILV secretion as nanosize EV at the CTL-target cell synaptic cleft, first described by Peters et al. [17]. Although the vesicles secreted by CTL were not referred at that time as exosomes, their formation and mode of exocytosis justifies such a nomenclature [6,17].

Subsequent publications demonstrated that T lymphocyte stimulation of T lymphoblasts (including CD4^+^ and CD8^+^ lymphocytes) with activation agonists produced non-directional secretion of nanosize EV (quoted as microvesicles) carrying pro-apoptotic FasL and Apo2L [11] via MVB-mediated degranulation [12], providing an alternative mechanism of TCR-controlled AICD that does not necessarily imply cell-to-cell contact [12,67]. Moreover, it was shown that upon TCR triggering T lymphoblasts secrete exosomes [67,68] containing TCR/CD3 [68], extending the early observations obtained in CTL forming synapses [17]. CTL MTOC reorientation is initially guided by a DAG gradient centered at the IS [69], generated by TCR-stimulated phospholipase C (PLC). DAG phosphorylation by diacylglycerol kinase α(DGKα) is involved in the spatiotemporal control of the DAG gradient [70,71] and MTOC polarization to the IS in CTL [69]. DAG activates, among others, several members of the PKC and PKD families [72], such as PKCδ, which is necessary for the polarization of lytic granules and cytotoxicity in mouse CTL [73,74].

### 2.2. Th Immune Synapse

Polarized secretion upon Th IS formation has been less studied than polarized CTL secretion [75]. Th IS are more stable and longer (from minutes up to several hours) than CTL IS (few minutes) [30,53]. Th IS are required for both directional and continuous cytokine secretion [30,53]. These cytokines are contained in secretory vesicles and IL-2, IFN-γ-containing secretory vesicles undergo polarized traffic to the F-actin poor area at cSMAC [49,53,54,76] as CTL lytic granules. Although the identity of the cytokine-containing secretory vesicles has not been characterized yet [75,77] they, most probably, are not MVB [77] (Figure 2).

Early reports in CD4^+^ Jurkat cells and T lymphoblasts demonstrated that stimulation with activation agonists [11] or anti-TCR [67,68] induced exosome secretion. Stimulation with a heterologous receptor agonist, that mimics TCR-derived signals leading to full T cell activation [78] and AICD [79], also induced exosome secretion in CD4^+^ Jurkat cells [67], suggesting that exosome secretion is a general consequence of T lymphocyte activation. IS formation by CD4^+^ Jurkat cells and superantigen-coated Raji B cells acting as an APC, which constitutes a well-established IS model [61,80,81], induces polarized MVB traffic towards the IS, MVB degranulation and exosome release [15,18] (Figure 2, right side panel). In this Th-APC IS model, a positive role of TCR-triggered DAG and its regulator DGKα [70], in polarized MVB traffic towards the IS was demonstrated [15,67,82]. As DGKα also controls late endosomes polarized traffic during invasive migration [83], and MTOC and lytic granules polarized traffic in CTL (described above), DAG and DGKα can be considered as general regulators of polarized traffic. DAG-activated PKCδ is needed for cortical actin reorganization at the IS, MTOC and MVB polarization to the IS and exosome secretion in this IS model [84]. Overall, this leads us to hypothesize that an altered actin reorganization at the IS may underlie the deficient MVB polarization occurring in PKCδ-interfered T cell clones [84]. In this model, DAG-activated PKD1/2 regulates MVB maturation and polarization leading to exosome secretion [59], suggesting that several regulatory points in exosome secretion are controlled by DAG.

Microvesicles or ectosomes budding from the Th cell plasma membrane and accumulating at the IS have been described [85] (Figure 2, right side panel). These shedding vesicles were enriched in TCR and capable to trigger B-lymphocyte signaling via pMHC-II stimulation [85,86]. Thus, it appears that distinct types of EV from Th lymphocytes are secreted at the IS. Further research will be necessary to establish whether these subtypes of EV trigger different Th effector responses or, on the contrary, redundantly or synergistically trigger the same responses.

## 3. B Lymphocyte-APC Immune Synapse: Antigen Capture and Processing by B Lymphocytes

B lymphocytes are capable of forming two classes of IS, both directing polarized traffic and secretion and being crucial to the development of adaptive immune responses: one for antigen capture and processing (Figure 3, upper panel), and the other one for antigen presentation to Th cells (Figure 3, lower panel) [33]. This subheading deals with the former IS.

Once BCR recognizes the antigen attached to the surface of specialized APC such as follicular dendritic cells (FDC), DC or macrophages, cell polarity is immediately established upon IS formation that seems to allow B lymphocytes to rapidly focus their antigen-extraction machinery to efficiently capture antigen [33]. B lymphocytes contain late endosomal compartments with MVB and lysosome characteristics called MHC-II-enriched (MHC-II^+^) compartments, involved in antigen processing and antigenic peptide binding to MHC-II molecules [9] (Figure 3). BCR stimulation with agonist antibodies, that mimic the activation of B lymphocytes by FDC in vivo, induces PKC-dependent formation and maturation of MHC-II^+^ compartments [87,88] and PKC- and PKD1/3-dependent pMHC-II^+^ exosome secretion [16,59,89], demonstrating a link between antigen processing/presentation and exosome secretion.

Remarkably, upon BCR binding to the antigen at the IS formed with APC such as FDC, B lymphocytes quickly polarize their MTOC, together with MHC-II^+^ compartments and secretory lysosomes (SL) towards the F-actin-poor cSMAC. MHC-II^+^ compartment and SL degranulation at the cSMAC allows localized secretion of lysosomal hydrolases [32,90], providing an acidic and protease-enriched, restricted extracellular area and favoring antigen extraction prior to internalization of BCR-bound antigen [32,90]. Subsequently, antigen-BCR complexes are internalized into late endosomal and lysosomal compartments in which the processed antigen and MHC-II molecules converge and pMHC-II complexes are assembled [33,91] (Figure 3, upper panel). In vivo B lymphocyte-FDC IS contains pMHC-II^+^ exosomes attached to the FDC surface. As FDC do not express MHC-II at the cell surface [92], these exosomes are, most probably, secreted by B lymphocytes upon synaptic MHC-II^+^ compartments degranulation concomitantly with antigen extraction. Supporting this possibility, purified B lymphocyte-derived exosomes can ex vivo efficiently bind to FDC [6,92]. Further experiments (i.e., direct evidence for MHC-II^+^ compartment degranulation at the B lymphocyte-FDC IS) will be necessary to formally demonstrate the B cell origin of these exosomes and to study the potential contribution of the exosomal proteases [93] to the antigen extraction process.

## 4. B Lymphocyte–Th Immune Synapse: Antigen Presentation and Exosome Secretion by B Lymphocytes

B lymphocyte surface pMHC-II complex interaction with Th TCR leads to the formation of the second class of B lymphocytes IS, which triggers Th-B lymphocyte bidirectional activation signals [33], enabling B lymphocytes to receive the activation signals to become fully activated [94]. Polarized secretion, including exosome secretion, is induced in the Th IS side (see subheading 2), and MTOC and MHC-II^+^ compartment polarization to the synaptic interface is induced by pMHC-II stimulation on the B lymphocyte side [94]. Most probably, this causes B lymphocyte exosome secretion upon Th help [16,89] (Figure 3, lower panel), although no direct data regarding MVB degranulation or the presence of B-derived exosomes at the B lymphocyte–Th synaptic cleft has been obtained yet. Thus, apart from the fact that BCR crosslinking increases MVB number and exosome secretion in B lymphocytes (subheading 3), pMHC-II crosslinking also induces exosome secretion in B lymphocytes [16]. The fact that pMHC-II on exosomes efficiently stimulates primed antigen-specific T cells [9,16] suggests a major role of B cell-derived exosomes as extracellular amplifiers of antigen presentation in vivo [91] (see below).

Table 1 summarizes the different types of IS and the differences and similarities in terms of exosome production by T and B lymphocytes, underlining the existing gaps and focusing on the most relevant points commented in the text.

## 5. Immune Regulation by B and T Lymphocyte-Produced Exosomes

Extensive and recent reviews have been published regarding the contribution of EV and exosomes to several immune responses [23,91,102] and concerning the role of EV in genetic information exchange among immune cells [19,103]. Thus, we will focus on some regulatory functions concerning certain immunologically relevant molecules present on the surface of B and T lymphocyte-derived exosomes, excluding those derived of the exosomal miRNA content, since excellent reviews have already dealt with this important subject [102,103,104,105].

### 5.1. Role of B-Lymphocyte Produced Exosomes in Antigen Presentation and Immunoregulation

Non-stimulated B lymphocyte-derived pMHC-II^+^ exosomes can directly prime T cells [9]. On the other hand, pMHC-II^+^ exosomes produced by Th-activated B lymphocytes (subheading 4) can induce further and maintained stimulation of antigen-specific Th lymphocytes [91]. The fact that pMHC-II on exosomes secreted from activated B lymphocytes efficiently stimulate primed (but not naïve) antigen-specific Th lymphocytes suggests a positive role of B-lymphocyte derived exosomes in the maintenance of an ongoing immune response [16]. Thus, apart from the initial Th activation by B lymphocyte plasma membrane pMHC-II complexes, an increased proportion of B lymphocyte pMHC-II is secreted in exosomes upon interaction with cognate Th lymphocytes, and these exosomes may travel and attach to other distant APC (DC, FDC), via exosomal antigen-BCR complexes binding to APC Fc receptors [106,107], and stimulate naïve Th lymphocytes resulting in amplification of Th activation [91,108] or maintenance of antigen-specific memory T lymphocytes [16]. FDC can display surface-anchored exosomes bearing native antigen and pMHC-II, enabling the simultaneous interaction of FDC with naïve antigen-specific B lymphocytes and pMHC-I-specific Th lymphocytes, which would lead to antigen-specific B-Th lymphocytes cooperation and to the development of B cell immunity [102].

Interestingly, a complex role of the different lymphocyte subsets (CD4^+^ T lymphocytes, CD8^+^ CTL, NK cells) and DC in the in vivo CTL immune response to exosomal antigen presented on B-cell derived exosomes have been described [109], which suggests an intricate interplay of cooperating lymphocytes for exosome-derived immunogenicity and emphasizes the role of APCs in the immune response to exosomes [109].

More recently (see below), it has been described that B-lymphocyte derived exosomes may have immunoregulatory functions which are independent of their ability to present antigen as nucletotidase enzymatic activity present on B lymphocyte-derived exosomes (see below) facilitates CTLs function suppression [110].

### 5.2. Role of T Lymphocyte-Produced Exosomes on Apoptosis and Immunoregulation

AICD [111] is a process by which over-activated T cells are eliminated, thus preventing potential autoimmune attacks, contributing to immune homeostasis and peripheral immune tolerance [66]. Fas-FasL interaction mediates AICD of T cell hybridomas [112,113,114] and mature normal T lymphocytes [64,115]. FasL^+^ exosomes/EV able to trigger Fas-induced apoptosis were first described as FasL^+^ APO2L^+^ microvesicles secreted upon activation agonist-stimulation of T lymphocytes [11,12], unveiling a new mechanism for the rapid induction of autocrine or paracrine cell death during immune regulation. T-cell produced microvesicles were shown to be canonic exosomes induced also upon TCR stimulation [67,68], to bear the TCR expressed by the producing T lymphocyte [68] and to trigger FasL-dependent ex vivo apoptosis of CD4^+^ Jurkat cells [67], suggesting these FasL^+^ T lymphocyte-produced exosomes may have an unsuspected antigen-specific immunoregulatory role in vivo, although this has not been formally demonstrated yet.

The presence of exosomes (carrying FasL in one of the reports) in human plasma, capable of ex vivo induce Fas-dependent apoptosis of Fas^+^CD4^+^ lymphocytes, supports an in vivo immunoregulatory role of FasL^+^ exosomes, although the cellular origin of these plasma exosomes was not established [116,117]. In line with this, it was found that CD8^+^ T-lymphocytes from ovalbumin (OVA)-specific, TCR-transgenic OT-I mice produced exosomes bearing OVA-specific TCR and FasL. These exosomes bound to DC via pMHC-I/TCR and CD54/LFA-1 interactions, lead to pMHC-I expression downregulation and ex vivo FasL/Fas interaction-dependent DC apoptosis [118]. In addition, CD4^+^ T-lymphocytes from OVA-specific, TCR-transgenic OT-II mice produced exosomes bearing OVA-specific TCR and FasL, that also bound to DC via pMHC-I/TCR and CD54/LFA-1 interactions, leading to in vivo CD8^+^ CTL response inhibition, although it was not established whether this inhibition was due to FasL/Fas interaction-mediated apoptosis induction [119]. These immunoregulatory roles of CD4^+^ and CD8^+^ T lymphocyte-derived exosomes were described in TCR transgenic mice. Thus, further experiments are necessary to extend the results to a more physiologic scenario.

Considering all these results, exosomes produced by CD4^+^ and CD8^+^ T lymphocytes bearing FasL appear to be capable of inducing apoptosis and/or regulate the functions of other immune cells in an antigen-specific manner. Supporting these findings, it has been shown that both ex vivo AICD and CTL-mediated cytotoxicity were decreased in T lymphoblasts from mice lacking PKD2, correlating with TCR-stimulated exosome secretion inhibition [59].

More recently, it has been demonstrated by using OVA-specific, TCR-transgenic OT-I mice that full stimulation of murine CTL with IL-12 induces functionally distinct CTL-derived exosomes, which can thereby activate naïve, bystander CD8^+^ lymphocytes without the presence of antigen [120]. In addition, the same lab has shown that fully stimulated CTLs can secrete exosomes that preferentially enhance the activation of low-affinity CTLs [121]. Thus, CTL-derived exosomes may contribute to the host’s ability to mount a robust immune response to one infection, while maintaining immunity against other pathogens via activation of bystander CD8^+^ lymphocytes through activated CTL-derived exosomes [120], which might also contribute to communication between fully activated, high-affinity CTL, and low-affinity CTL [121].

T regulatory cells (Treg) are important immunoregulatory T lymphocytes involved in the prevention of inflammatory damage produced after infection, autoimmunity, or allergy [122], and have been shown to release, upon activation, more exosomes per cell than any other T lymphocyte subpopulation [123]. Treg suppresses pathogenic Th1 lymphocyte responses through gene silencing mediated by microRNA-containing exosomes. When Treg-mediated exosome release was inhibited (i.e., Rab27 KO) there was a suppression of the ability of Treg to suppress Th1 lymphocyte proliferation in vivo and prevent disease [123]. Moreover, Treg suppresses effector T lymphocytes through exosomal CD39 and CD73 ectonucleotidases, which catalyze the production of extracellular adenosine [124,125]. Treg-derived exosomes induce naïve T lymphocyte differentiation into Treg, and kidney allograft survival prolongation [126], suggesting that Treg-derived exosomes may represent a promising new therapy for transplant tolerance induction [127]. EV enriched in exosomes released from Treg contained specific miRNAs and iNOS which, once delivered to T lymphocytes, inhibited T lymphocyte alloreactivity by perturbing cell cycle progression, inducing apoptosis, and converting target T lymphocytes into Treg [126]. All these pieces of evidence support that exosome secretion plays an important role in Treg-mediated suppression. Exosomes may offer some advantages compared to their parent cells as a therapy to induce immune tolerance in transplantation: they are fully cell-free, are stable after in vivo infusion and can be easily stored [126].

All these results open the possibility that antigen-specific CD4^+^ and CD8+ T lymphocytes and Treg, can modulate immune responses via exosomes. Thus it was hypothesized that exosomes may be useful for the treatment of certain autoimmune diseases [118,119]. In this context, Treg-derived exosomes have been shown to inhibit the development of murine arthritis and to prevent certain autoimmune diseases [128]. The clinical value of Treg-derived exosomes has been extensively discussed in several recent reviews [128,129].

## 6. Some Potential Clinical Applications of Immune Cells-Derived Exosomes

Many recent reviews dealing with the potential use of exosomes produced by different cell types as biomarkers of several pathologies and therapies have been published [130,131,132], thus we will center only on some recent publications concerning either preclinical studies with exosomes or potential therapeutic approaches related to exosomes produced by B and T lymphocytes.

Although the exact content of T lymphocyte-derived exosomes and microvesicles remains to be determined (subheading 1.1), in the initial studies it was hypothesized that some of the molecules present in exosomes may provide these EV with important biological functions. The potential effector functions of both B and T lymphocyte-derived exosomes most probably include antigenic specificity since they display either pMHC-II capable of activating antigen-specific Th lymphocytes (B lymphocyte-produced exosomes) or specific TCR capable of recognizing pMHC (T lymphocyte-derived exosomes). For instance, it was early suggested that TCR/CD3 on T-lymphocyte-derived exosomes [17,68] would allow them to specifically deliver signals to cells bearing TCR-recognized pMHC [17]. In addition, T lymphocyte-derived exosomes contain pro-apoptotic molecules such as perforin/granzyme [17] and FasL and Apo2L [12,67] (see above). Although the role of these exosomal molecules in both physiologic and pathologic biological responses in vivo has not been fully established (subheading 5), several strategies have been developed considering the presence of TCR and pro-apoptotic molecules on exosomes would endow them with both antigenic specificity and cytotoxicity. This assumption has been recently validated in preclinical studies by using exosomes derived from Chimeric Antigen Receptors (CAR)-T lymphocytes [133]. CAR consist of an extracellular domain, which confers antigen-recognition specificity, a transmembrane domain, and an intracellular signaling domain, which provides activation signals to T lymphocytes [134]. Exosomes released from CAR-T lymphocytes exhibit excellent potential for use as direct attackers in immunotherapy, and the authors demonstrate in their study that ex vivo-produced human exosomes carrying human EGFR and HER2-specific CAR have potent in vivo activity against human tumor in xenograft models [133]. The use of CAR exosomes as cell-free immunotherapy has advantages, such as their independence of CAR-T lymphocytes life span and replication, their stability, the low risk of side toxicity (i.e., cytokine release syndrome) when compared with CAR-T lymphocytes, and the fact that exosomes lacking PD-1 (in contrast to PD-1-expressing T lymphocytes) are refractory to PD-L1 immunosupression by the tumor [133]. In addition, exosomes could be delivered through the blood circulation and other biological fluids as evidenced by the abundance of exosomes found in most biological fluids. Moreover, exosomes have the ability to cross biological barriers such as the blood–brain barrier and blood-tumor barrier as corroborated by the extensive presence of tumor cell-derived exosomes in bodily fluids [135]. Considering that it is not known whether CAR signaling or the exosomal proapoptotic molecules mediate the therapeutic effects observed, these findings require further validation, although they strongly suggest the use of exosomes as biomimetic nanovesicles in the therapy against tumors [133].

The presence of biologically active molecules, such as TNFα, in exosomes derived from bystander cells different to T and B lymphocytes, might also affect the immune response, and several TNF family proteins have been identified on exosomes. It is interesting to remark that exosomes produced by synovial fibroblasts from patients developing rheumatoid arthritis (but not from healthy controls) contain membrane-bound TNFα [136] CD4^+^ T lymphocytes are a major cell population in the inflammatory infiltrate of rheumatoid arthritis patient joints. CD4^+^ T lymphocytes co-cultured with these exosomes, develop apoptosis resistance, and blockade of exosomal TNFα may lead to partial restoration of CD4^+^ T lymphocyte AICD [136], ameliorating the disease. In addition, the identification of specific pathways that regulate the induction of exosomes, not only by T and B lymphocytes, may lead to novel therapeutic targets for rheumatoid arthritis treatment [136]. However, it should be taken into account that in rheumatoid arthritis patients there are exosomes in the synovial fluid of inflamed joints, which originate from synovial fibroblasts and different infiltrated cells including platelets, granulocytes, monocytes, neutrophils, mesenchymal stem cells, and T and B lymphocytes [137]. Synovial exosomes, either by themselves or by inducing synovial fibroblasts, can cause joint inflammation and destruction mainly through extracellular matrix degradation, alteration in the cell-to-cell communication, and triggering of inflammation and autoimmune responses [137]. On the contrary, mesenchymal stem cells-derived exosomes can decrease cartilage degradation and joint destruction through antifibrotic, anti-apoptotic, and immunomodulatory effects, as well as suppressing synoviocyte proliferation and promoting cartilage regeneration in rheumatic diseases [137], providing an alternative therapeutic approach for the rheumatoid arthritis treatment.

In the same context of autoimmune diseases, it has been shown that circulating exosomes from patients suffering from systemic lupus erythematosus (SLE) induce a pro-inflammatory immune response. SLE exosomes elicit a significant inflammatory response (pro-inflammatory cytokines) on peripheral blood lymphocytes (PBL) in a Toll-like receptor (TLR)-dependent manner [138]. Although in these studies the cellular origin of the pro-inflammatory exosomes was not defined, binding of these exosomes to the cell surface, but also to endosomal, TLR on PBL led to TLR-mediated pro-inflammatory cytokine secretion [138]. Given the strong immunostimulatory effect of SLE exosomes, the removal of the circulating exosomes might offer a new therapeutic approach to SLE [138].

In summary, several pieces of evidence indicate that exosomes, not only the ones produced by T and B lymphocytes, play important roles in the immunomodulation and are associated with the immune pathogenesis of autoimmune diseases, including rheumatoid arthritis, Sjogren’s syndrome, systemic lupus erythematosus [139], and systemic sclerosis [140]. Please refer to some excellent and recent reviews regarding the role of exosomes in several autoimmune diseases and some potential therapeutic approaches [137,139,140].

Recently it has been described that B lymphocyte-derived exosomes may have immunoregulatory functions that are independent of their ability to present antigen since certain enzymatic activities of B lymphocyte-derived exosomes facilitate the suppression of CTL function [110]. It has been shown that B-lymphocyte derived EV (most probably exosomes) carrying ectonucleotidases CD39 and CD73 hydrolyzed ATP released from chemotherapy-treated tumor cells into adenosine [110]. It is well established that extracellular adenosine inhibits T lymphocyte function and CTL activity directed against tumor cells by binding to T lymphocyte A2A adenosine receptors. Therefore, ectonucleotidases released on B lymphocyte-derived exosomes attenuate chemotherapeutic efficacy by inhibiting CD8^+^ T lymphocyte responses [110,141]. Silencing Rab27a in B lymphocytes inhibited exosomal nucleotidase production and improved the antitumor effect of chemotherapy [110,141]. Thus, decreasing exosome secretion by B lymphocytes may be used as a therapeutic approach during the treatment of certain tumors.

## 7. Conclusions

During the last 15 years, exosomes have been studied as new biological entities involved in intercellular communication during a multitude of physiological and pathological processes, in particular in the context of the immune system. In addition, cumulative pieces of evidence demonstrate that exosomes constitute promising biomarkers for both diagnosis and prognosis in several relevant pathologies (i.e., cancer, autoimmune disorders, etc.). However, it is necessary to improve the existing experimental approaches, but also the characterization techniques, to study exosomes, to surely establish the role of exosomes in these pathologies and to distinguish exosome functions from other EV functions. In this context, the development of existing, but also new, nanotechnology and imaging tools for the analysis and characterization of exosomes will indeed allow addressing the contribution of exosomes to these pathologies. In addition, a better understanding of the signals involved in MVB maturation and traffic will allow designing strategies to modulate exosome secretion and hence modify their function. Cell-free, exosome-based immunotherapeutic strategies have indeed a promising future.

## Figures and Tables

**Figure 1 ijms-21-02631-f001:**
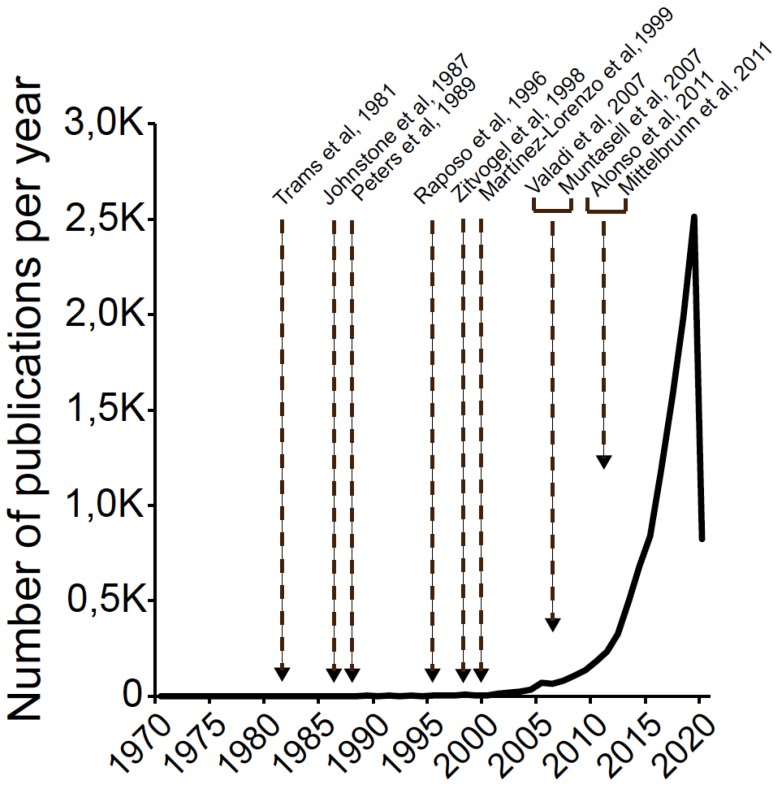
Exosome timeline and publications. A search was performed in PubMed on 2 April 2020 to find, for each year of publication, articles using the given term “exosomes” and the related term “small extracellular vesicles” as text word. Data are not normalized to the total number of biology and biomedicine research publications. Arrows on the graph indicate the year of publication of some milestone papers mentioned in the text.

**Figure 2 ijms-21-02631-f002:**
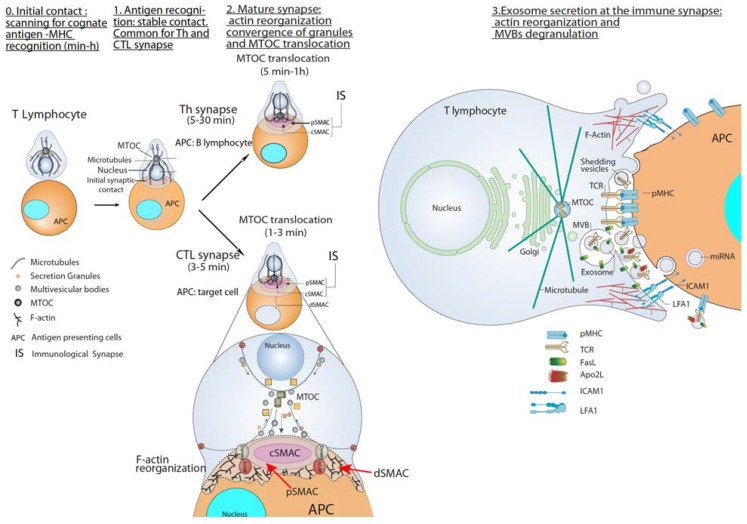
T lymphocyte—antigen-presenting cells (APC) immune synapse (IS) and polarized secretion. Stages 0 and 1 are common for both Th and cytotoxic T lymphocytes (CTL) IS. After the initial scanning contact of TCR with pMHC on APC, Th effector T lymphocytes (upper panel) form mature IS with antigen-presenting B lymphocytes within several minutes. This IS lasts many hours during which de novo cytokine (i.e., IL-2, IFN-γ) production and secretion occur, which require continuous T-cell receptors (TCR) signaling. Primed effector CTL (lower panel) establish more transient, mature IS after scanning their target cells (i.e., a virus-infected cell), and deliver their lethal hits within a few minutes. Secretory lysosomes (lytic granules) are very rapidly transported (within very few minutes) towards the microtubule organization center (MTOC) (in the minus “–“ direction) and, almost simultaneously, the MTOC polarizes towards the central supramolecular activation complex (cSMAC) of the IS, an F-actin poor area that constitutes a secretory domain. MTOC translocation to the IS appears to be dependent on dynein anchored to the Adhesion and Degranulation Promoting Adapter Protein (ADAP) at the peripheral SMAC (pSMAC), which pulls MTOC in the minus direction. In both types of IS (lower zoom panel), the initial F-actin reorganization in the cell-to-cell contact area, followed by a decrease in F-actin at the cSMAC and an accumulation at the distal SMAC (dSMAC) appears to be involved in granule secretion. In stage 3, MVB fusion with the plasma membrane occurs in both types of IS and leads to TCR-containing exosome polarized secretion at the IS. The exosomes released in Th IS contain proapoptotic FasL and Apo2L and can induce target cell death or Th cell death (AICD). TCR–containing shedding microvesicles have been described in Th IS.

**Figure 3 ijms-21-02631-f003:**
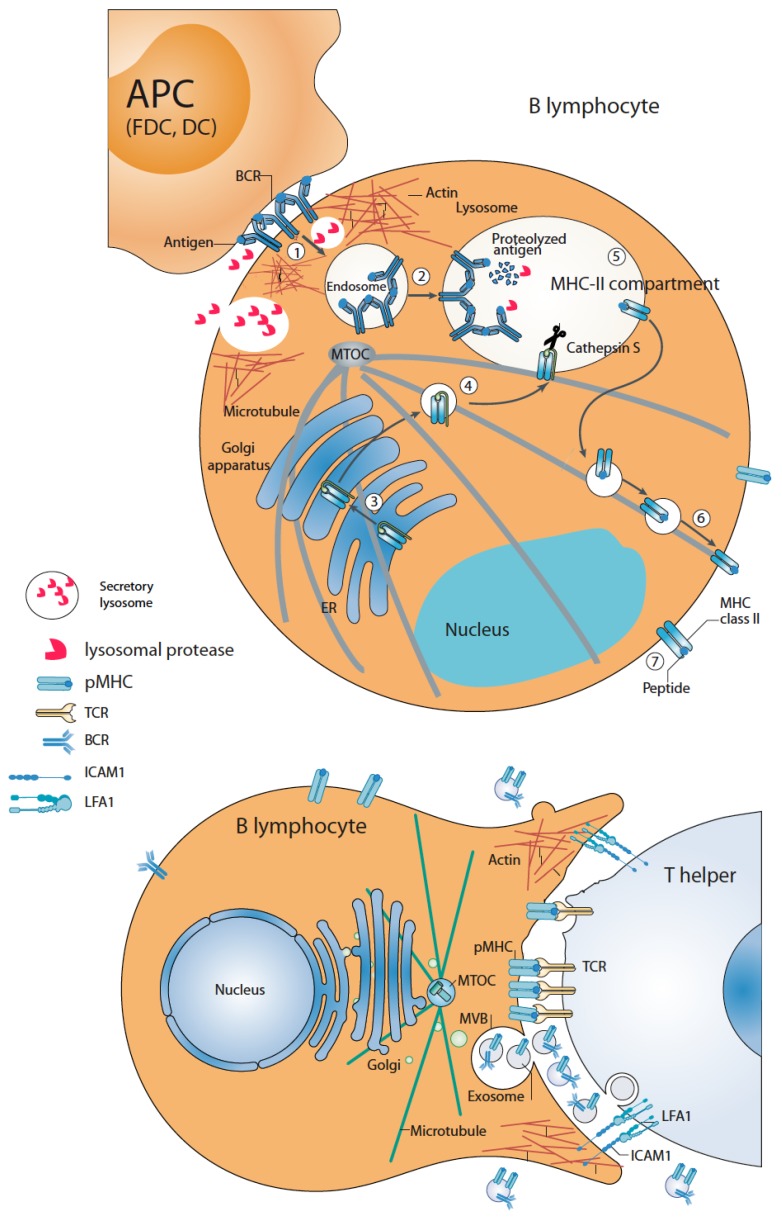
B lymphocytes form two classes of IS: antigen-capture/processing (upper panel) and antigen presentation to Th lymphocytes (lower panel). Upper panel, antigen recognition on APC is mediated by B-cell receptors (BCR). BCR triggering induces actin reorganization at the IS and MTOC and secretory lysosomes recruitment towards the IS and protease secretion at the extracellular synaptic cleft, facilitating antigen processing and extraction. Subsequently, antigen-BCR complexes are extracted and endocytosed in a clathrin-dependent process to early endosomes (stage 1) and, then to late endosomal, MHC-II^+^ compartments (stage 2) where antigen and MHC-II molecules trafficking from Golgi (stages 3 and 4) converge. Coordinately with antigen endocytosis, antigen-BCR interaction promotes the biogenesis of this compartment to facilitate antigen processing. In this compartment, the antigen is additionally processed by proteases to form MHC-II/antigenic peptide complexes (pMHC-II) (stage 5) that, subsequently, are exported and distributed to the cell surface (stages 6 and 7). Lower panel, pMHC-II complexes are recognized by Th lymphocyte TCR forming a second secretory IS. This leads to a polarized phenotype leading to MTOC and MHC-II^+^ compartment (MVB) polarization. Local secretion of exosomes containing pMHC-II complexes at the B lymphocyte IS side is represented (see text for further details).

**Table 1 ijms-21-02631-t001:** Different IS and polarized secretory traffic in T lymphocyte IS and B lymphocyte IS.

	CTL/APC	Th/APC	B/APC	B/Th
Exosome secretion	+ [17]	+ [15]	Unknown ^1^	+ [16]
MTOC polarization	+ [45]	+ [95]	+ [90]	+ [94]
Secretory granules polarization	Lytic granules, MVB [29]	Lymphokine-containing granules, MVB [15,49]	SL, MHC-II^+^ compartment [90]	MHC-II^+^ compartment [94]
DAG/DGK-control of MTOC polarization	+ [69]	+ [15,69]	Unknown	Unknown
PKC/PKD control of secretory granules/MVB traffic	+ (PKCδ, PKCθ) ^2^ [34,73,96]	+ (PKCδ) [84] (PKD1/2) [59]	+ (PKCζ) [88,90] (PKD1/3) [59]	Unknown
F-actin reorganization at IS	+ (CDC42/WASP/ARP2/3, Formins: FMNL1, Dia1) [31]	+ (CDC42/WASP/ARP2/3) (Formins: FMNL1, Dia1) [49,97]	+ (CDC42/WASP/ARP2/3) Ezrin, Moesin) [32]	Unknown (TAGNL2?) [98]
F-actin reduction at cSMAC and MTOC polarization	+ [46]	+ (PKCδ dynein) [50,51,52,84]	+ (dynein, proteasome) [99,100]	Unknown
F-actin reduction at MTOC and MTOC polarization	Unknown	Unknown	+ (Proteasome) [100,101]	Unknown

^1^ Formal proof has not been obtained, although indirect evidence exists. ^2^ Brackets indicate the molecular components involved.

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
