# Peer review of "Inducible Polarized Secretion of Exosomes in T and B Lymphocytes"

_ijms, 2020, doi:10.3390/ijms21072631_

Round 1

Reviewer 1 Report

In this review entitlet “Inducible polarized secretion of exosomes in T and B lymphocytes“ by Victor Calvo and Manuel Izquierdo authors summerize studies that define exosomes biology, considering in particular the events leading to polarized exosome secretion at the immune synapsis. Moreover they analyze some of the therapeutic applications of exosomes produced by immune cells. Overall I consider this review interesting, suggesting novel interpretations and increase our knowledge. However, there are few points that need to address:

  1. At the beginning of the manuscript authors decide to dedicate a chapter to “normalization attempts and isolation protocols”. I don't think it is needed to the aim of the review, differently it is off topic. I think it could be summerize in the first chapter.

  2. At the end of the manuscript authors suggest potential therapeutic approaches. I think that they should go deeper in this topic and mention some studies and reviews already avaible in literature, particularly regarding autoimmune diseases:

    (i.e. Authors should mention

    1. Zhang HG et al. Membrane form of TNF-alpha presented by exosomes delays T cell activation-induced cell death. J. Immunol. 2006;

    2. Lee, JY et al. Circulating exosomes from patients with systemic lupus erythematosus induce an proinflammatory immune response. Arthritis Res. Ther. 2016;

    3. Colletti M, et al. Exosomes in Systemic Sclerosis: Messengers Between Immune, Vascular and Fibrotic Components? Int J Mol Sci. 2019;

    4. Tan Let al. Recent advances of exosomes in immune modulation and autoimmune diseases. Autoimmunity. 2016;

    5. Tavasolian F, et al. Exosomes: Effectual players in rheumatoid arthritis. Autoimmun Rev. 2020 Mar 12:102511).

Author Response

Reviewer 1

In this review entitled “Inducible polarized secretion of exosomes in T and B lymphocytes“ by Victor Calvo and Manuel Izquierdo authors summerize studies that define exosomes biology, considering in particular the events leading to polarized exosome secretion at the immune synapsis. Moreover they analyze some of the therapeutic applications of exosomes produced by immune cells. Overall I consider this review interesting, suggesting novel interpretations and increase our knowledge. However, there are few points that need to address:

 Thank you very much for the positive comments. We acknowledge the improvement suggestions.

  1. At the beginning of the manuscript authors decide to dedicate a chapter to “normalization attempts and isolation protocols”. I don't think it is needed to the aim of the review, differently it is off topic. I think it could be summerize in the first chapter.

Referee is right. The other referee raised the same point. Thus, we have deleted the section 1.2 and we have summarized some of its content as “relevant facts” in exosome research in a sentence added to the first section (1.1).

  1. At the end of the manuscript authors suggest potential therapeutic approaches. I think that they should go deeper in this topic and mention some studies and reviews already avaible in literature, particularly regarding autoimmune diseases:

(i.e. Authors should mention

  1. Zhang HG et al. Membrane form of TNF-alpha presented by exosomes delays T cell activation-induced cell death. J. Immunol. 2006;
  2. Lee, JY et al. Circulating exosomes from patients with systemic lupus erythematosus induce an proinflammatory immune response. Arthritis Res. Ther. 2016;
  3. Colletti M, et al. Exosomes in Systemic Sclerosis: Messengers Between Immune, Vascular and Fibrotic Components? Int J Mol Sci. 2019;
  4. Tan Let al. Recent advances of exosomes in immune modulation and autoimmune diseases. Autoimmunity. 2016;
  5. Tavasolian F, et al. Exosomes: Effectual players in rheumatoid arthritis. Autoimmun Rev. 2020 Mar 12:102511).

We have further developed this topic, thus we have now included some sentences in this regard and associated references, as requested. We would like to remark that, in a relevant proportion of these papers, the cells producing the exosomes are neither T nor B lymphocytes, and thus this topic remains slightly out of the scope of this review. However, we have followed referee’s request considering that the mechanisms controlling MVB traffic and exosome secretion appear to be conserved among different cell types.

Reviewer 2 Report

This review article aims to gather together information on polarized secretion of exosomes in T and B cells and the related biological functions. Overall, I found the review well organized and the sections on exosome released from T- and B- cells relatively clear. References and the literature are quite updated and the figures are excellent. I have several suggestions to improve the quality of this review article.

The manuscript contains many typos and grammar mistakes, and words are sometimes missing. For instance, line 100and 101: the words “on” and “of” are missing. The sentence L 107-112 is far too long and there is one “and” missing. Please split the sentence. Grammar and style editing is required.

On several occasions, the term “F-actin-depleted area” (for cSMAC) is used while super resolution microscopy has shown that a fine network of actin exists at the cSMAC and is critically involved in secretion. This old term can be misleading for the readers. I think the authors should be more specific/accurate on this aspect, which is highly relevant to their focus. In this regard, a recent review article has addressed in a very detailed manner actin remodeling at the immunological synapse (Wurzer et al. Cells 4, 463).

All abbreviations should be checked since they are sometimes used before being defined, e.g. L 117.

Line 40 – 41 “(…) and since then this proposal has been widely accepted by the scientific community although not systematically standardized yet (see below)”; I don’t believe this comment is necessary. It leads to section 1.2 that in my opinion is completely unnecessary and out of context for this review.

Figure 1 “Exosome timeline and publications”; the authors should also consider the term small extracellular vesicles (small EVs) in the PubMed research. Indeed, from 2018 ISEV – also cited by the authors – it has been stressed the importance of nomenclature and several authors changed the term exosomes for small EVs in their publications.

Section 1.3; the author already mentioned some basic information about exosomes release in the introduction. The most relevant information for this review is the presence of exosomes in MVBs, which are coordinated by microtubules towards the IS. The whole exosome biogenesis and composition section is wildly described in numerous reviews and the main topic of a review cannot be touch only starting in section 4. Thus, I would strongly suggest to the authors to delete this section and focus their attention on the main goal of the review. Polarization of exosomes towards the IS is already an advanced topic, the reader is probably already aware about basic information regarding exosomes.

Author Response

Reviewer 2

This review article aims to gather together information on polarized secretion of exosomes in T and B cells and the related biological functions. Overall, I found the review well organized and the sections on exosome released from T- and B- cells relatively clear. References and the literature are quite updated and the figures are excellent. I have several suggestions to improve the quality of this review article.

 Thank you very much for the positive comments. We acknowledge the improvement suggestions.

The manuscript contains many typos and grammar mistakes, and words are sometimes missing. For instance, line 100and 101: the words “on” and “of” are missing. The sentence L 107-112 is far too long and there is one “and” missing. Please split the sentence. Grammar and style editing is required.

Referee is right. Accordingly, we have now extensively edited the manuscript and corrected these mistakes. Enclosed you will find a PDF copy tracking all the changes.

On several occasions, the term “F-actin-depleted area” (for cSMAC) is used while super resolution microscopy has shown that a fine network of actin exists at the cSMAC and is critically involved in secretion. This old term can be misleading for the readers. I think the authors should be more specific/accurate on this aspect, which is highly relevant to their focus. In this regard, a recent review article has addressed in a very detailed manner actin remodeling at the immunological synapse (Wurzer et al. Cells 4, 463).

Referee is right, although some recent papers (published in 2019) use this term to define this area at the cSMAC, recent papers (and the mentioned review) further clarify this issue. Indeed, this region is referred more accurately as an “actin poor” or “actin hypodense” region, since it contains a fine F-actin network, which can only be visualized using super resolution imaging. Accordingly, we have now changed the adjective “depletion” by “reduction” when referred to F-actin at the cSMAC, for clarity’s sake.
We have also included a sentence at section 1. 2 clarifying this point and some related references. We acknowledge the referee for this important point.

All abbreviations should be checked since they are sometimes used before being defined, e.g. L 117.

Referee is right; we have checked and corrected abbreviations when necessary.

Line 40 – 41 “(…) and since then this proposal has been widely accepted by the scientific community although not systematically standardized yet (see below)”; I don’t believe this comment is necessary. It leads to section 1.2 that in my opinion is completely unnecessary and out of context for this review.

Referee is right. This sentence has been now deleted. In addition, we have deleted section 1.2, as requested also by the other referee (see above), and we have summarized its content since it contains some relevant facts for exosome understanding into a new sentence added to section 1.1.

Figure 1 “Exosome timeline and publications”; the authors should also consider the term small extracellular vesicles (small EVs) in the PubMed research. Indeed, from 2018 ISEV – also cited by the authors – it has been stressed the importance of nomenclature and several authors changed the term exosomes for small EVs in their publications.

We have now performed an additional search considering the term “small extracellular vesicles” (Advanced Search in Pubmed) and retrieved 244 records. This is a small number of records, when compared with the records derived from the “exosome” search (11442), and did not significantly change the graphic. We have now included these records and actualized the search date to April 2020.

Section 1.3; the author already mentioned some basic information about exosomes release in the introduction. The most relevant information for this review is the presence of exosomes in MVBs, which are coordinated by microtubules towards the IS. The whole exosome biogenesis and composition section is wildly described in numerous reviews and the main topic of a review cannot be touch only starting in section 4. Thus, I would strongly suggest to the authors to delete this section and focus their attention on the main goal of the review. Polarization of exosomes towards the IS is already an advanced topic, the reader is probably already aware about basic information regarding exosomes.

Referee is right and we have now focussed our review in section 1.4. Accordingly, we have deleted the section 1.3. We have summarized some of its essential content as “relevant facts” in exosome research into the section 1.1.

Round 2

Reviewer 1 Report

The manuscript has been significantly improved and now warrants publication.

Reviewer 2 Report

I'm fine with the revisions.